# Appropriate Delivery Timing in Fetuses with Fetal Growth Restriction to Reduce Neonatal Complications: A Case—Control Study in Romania

**DOI:** 10.3390/jpm13040645

**Published:** 2023-04-08

**Authors:** Daniela-Loredana Bujorescu, Adrian Ratiu, Cosmin Citu, Florin Gorun, Oana Maria Gorun, Doru Ciprian Crisan, Alina-Ramona Cozlac, Ioana Chiorean-Cojocaru, Mihaela Tunescu, Zoran Laurentiu Popa, Roxana Folescu, Andrei Motoc

**Affiliations:** 1Doctoral School, “Victor Babes” University of Medicine and Pharmacy Timisoara, 2 Eftimie Murgu Square, 300041 Timisoara, Romania; 2Department of Obstetrics and Gynecology, “Victor Babes” University of Medicine and Pharmacy Timisoara, 2 Eftimie Murgu Square, 300041 Timisoara, Romania; 3Department of Obstetrics and Gynecology, Municipal Emergency Clinical Hospital Timisoara, 22–24 16 December 1989 Street, 300172 Timisoara, Romania; 4Cardiology Department, “Victor Babes” University of Medicine and Pharmacy, 2 Eftimie Murgu Square, 300041 Timisoara, Romania; 5MedLife Genesys Hyperclinic, 3 Dr. Cornel Radu Street, 310329 Arad, Romania; 6Neonatology Clinic, Municipal Emergency Clinical Hospital Timisoara, 22–24 16 December 1989 Street, 300172 Timisoara, Romania; 7Department of Balneology, Medical Recovery and Rheumatology, Family Discipline, Center for Preventive Medicine, Center for Advanced Research in Cardiovascular Pathology and Hemostaseology, “Victor Babes” University of Medicine and Pharmacy Timisoara, 2 Eftimie Murgu Square, 300041 Timisoara, Romania; 8Department of Anatomy and Embryology, “Victor Babes” University of Medicine and Pharmacy, 2 Eftimie Murgu Square, 300041 Timisoara, Romania

**Keywords:** premature birth, fetal growth restriction, ultrasonography, newborn complications

## Abstract

(1) Background: The main challenge in cases of early onset fetal growth restriction is management (i.e., timing of delivery), trying to determine the optimal balance between the opposing risks of stillbirth and prematurity. The aim of this study is to determine the likelihood of neonatal complications depending on the time of birth based on Doppler parameters in fetuses with early onset fetal growth restriction; (2) Methods: A case–control study of 205 consecutive pregnant women diagnosed with early onset FGR was conducted at the Obstetrics Clinic of the Municipal Emergency Hospital in Timisoara, Romania; The case group included newborns who were delivered at the onset of umbilical arteries absent/reversed end-diastolic flow, and the control included infants delivered at the onset of reversed/absent ductus venosus A-wave. (3) Results: The overall neonatal mortality rate was 2.0%, and there was no significant statistical difference between the two study groups. In infants delivered up to 30 gestational weeks, grades III/IV intraventricular hemorrhage and bronchopulmonary dysplasia were statistically significantly more frequent in the control group. Moreover, univariate binomial logistic regression analysis on fetuses born under 30 gestational weeks shows that those included in the control group are 30 times more likely to develop bronchopulmonary dysplasia and 14 times more likely to develop intraventricular hemorrhage grades III/IV; (4) Conclusions: Infants delivered according to the occurrence of umbilical arteries absent/reversed end-diastolic flow are less likely to develop intraventricular hemorrhage grades III/IV and bronchopulmonary dysplasia.

## 1. Introduction

Fetal growth restriction (FGR) is a common pregnancy complication and is one of the leading causes of stillbirth, neonatal mortality, and short- and long-term neonatal morbidity worldwide [1]. FGR affects lung development, leading to increased respiratory morbidity as well as increased duration of mechanical ventilation in premature infants delivered earlier than 32 weeks of gestation [2,3]. Therefore, intrauterine growth restriction increases the risk of both respiratory distress syndrome, characterized by progressive respiratory failure immediately after birth, and bronchopulmonary dysplasia (BPD), which is associated with chronic lung disease in adulthood [2,3]. Another adverse outcome for which infants with FGR are at increased risk is intraventricular hemorrhage. FGR is broadly classified, based on gestational age at the time of diagnosis, into early onset (<32 weeks) and late-onset (≥32 weeks) FGR. The differences between these two FGR phenotypes in terms of severity, natural history, Doppler findings, association with hypertensive complications, placental findings, and management constitute the basis for this classification [4,5].

Early onset FGR has a prevalence of 0.5%–1%, is usually more severe, and is more likely to be associated with abnormal umbilical artery Doppler than late-onset FGR. The underlying placental pathology is frequently similar to that seen in early onset preeclampsia (maternal vascular malperfusion), which explains the strong association of early onset FGR with preeclampsia. Early onset FGR is usually easier to detect compared with late-onset FGR, and the natural history tends to follow a predictable sequence of Doppler changes in the umbilical artery and ductus venosus. Therefore, in contrast to late-onset FGR, where the primary issue is the diagnosis, the main challenge in early onset FGR is management (timing of delivery), trying to determine the optimal balance between the opposing risks of stillbirth and prematurity. This study aims to ascertain the likelihood of neonatal complications depending on the time of birth in fetuses with early onset FGR based on Doppler parameters, taking into account that in early onset FGR, the time of delivery is primarily chosen based on Doppler parameters, weighing the risks and benefits to the infant.

## 2. Materials and Methods

### 2.1. Study Design and Settings

A case–control study of 205 consecutive pregnant women diagnosed with early onset FGR was conducted at the Obstetrics Clinic of the Municipal Emergency Hospital in Timisoara between 2018 and 2022. Participants were divided into two groups (cases and control) according to the management (delivery moment) of FGR. Newborns were followed up until discharge from the hospital to monitor them for the onset of early complications. The study was approved by the Ethics Committee of the Municipal Emergency Hospital Timisoara (No. I-15505/15.06.2020). This study was performed according to the STROBE guidelines [6].

### 2.2. Participants

All participants were diagnosed with early onset FGR according to the following criteria: estimated fetal weight (EFW) < 3rd percentile; gestational age less than 32 weeks at the time of diagnosis; and abnormal Doppler evaluation on umbilical arteries (UA), middle cerebral artery (MCA), or ductus venosus (DV). All infants included were delivered based on Doppler parameters.

The allocation of the groups (case and control group) was based on the Doppler parameter used to determine the delivery.

The inclusion was as follows: 

Case group: (1) infants diagnosed with early onset FGR; (2) infants delivered at onset of UA—reversed end-diastolic flow (REDF) (in fetuses under 30 weeks gestation) or onset of UA—absent end-diastolic flow (AEDF) (in fetuses between 30 and 34 weeks gestation).

Control group: (1) infants diagnosed with early onset FGR; (2) infants delivered at onset of absent or reversed DV a-wave (in fetuses under 30 weeks gestation) or onset of UA—REDF (in fetuses between 30 and 34 weeks gestation).

The cases that met the absolute delivery indications at the time of diagnosis were excluded: maternal status (severe preeclampsia, eclampsia, HELLP syndrome), repetitive fetal heart rate (FHR) decelerations, sinusoidal tracing, absent FHR variability with late decelerations, biophysical profile score (BPP) < 4. Cases with these pathologies were excluded because they required immediate delivery independent of Doppler parameters.

### 2.3. Variables, Data Sources, and Measurement

Data were extracted by two researchers from patients’ electronic medical records using a standardized data collection form. Establishing the diagnosis, all ultrasounds during pregnancy and the Doppler studies were performed by one certified, not blind, maternal-fetal medicine specialist (A.R.). The ultrasound machines used were GE Voluson E10 and GE Voluson E8. Gestational age was established by first trimester ultrasound. Estimated fetal weights and percentiles were calculated using Hadlock’s formula based on the biparietal diameter, head circumference, abdominal circumference, and femur length [7]. Doppler measurements were obtained from the umbilical arteries (UA) and ductus venosus (DV) in accordance with uniform standards [8].

Abnormal CTGs were characterized by fixed FHR baselines, loss of FHR variability, absence of accelerations, and presence of decelerations. A biophysical profile score < 6 was considered abnormal.

All participants were delivered by cesarean section.

The primary outcomes of interest were early complications among infants with early onset FGR. Two neonatology specialists have established the diagnosis of fetal complications. IVH was diagnosed using cranial ultrasonography. Findings of IVH were graded as: (1) grade I—bleeding was limited to the lining of the ventricles; (2) grade II—blood spills into the ventricles, but without ventricular dilation; (3) grade III—IVH with ventricular dilation; (4) grade IV—persistent intraparenchymal lesions. The diagnosis of BPD was established according to the National Institute of Child Health and Human Development (NICHD) workshop [9]. The diagnosis of necrotizing enterocolitis was established on the basis of a series of plain abdominal X-ray images, including anteroposterior and left lateral decubitus images. The signs of necrotizing enterocolitis were the finding of dilated bowel loops, intestinal pneumatosis, and portal venous air.

### 2.4. Statistical Analysis

Statistical calculations were performed using R version 4.2.0: A language and environment for statistical computing (R Foundation for Statistical Computing, Vienna, Austria. https://www.R-project.org/) and SPSS 20.0 software (SPSS Inc., Chicago, IL, USA). Continuous variables were presented as median (interquartile range) and compared by Mann–Whitney test. Categorical variables were expressed in count and percentage and were compared using Fisher’s exact test. Binomial logistic regression was performed to assess the independent predictive value of timing of delivery in the occurrence of early neonatal complications.

## 3. Results

### 3.1. Characteristics of Pregnant Women with Early Onset FGR

A total of 205 pregnant women with early onset FGR were included in this study. The mean age was 29.48 years, with no statistical difference between the control and case groups. Moreover, there was no statistical difference between groups according to height, weight, gestation, and parity. The median gestational age (GA) at diagnosis was 31 weeks of gestation (WG), and the median GA at delivery was 32 WG (Table 1). 

### 3.2. Fetal Doppler Findings

The most common Doppler abnormality in fetuses with early onset FGR was absent end-diastolic flow (AEDF) in an umbilical artery, which was also significantly more common in those included in the case group. Additionally, abnormal cardiotocography (CTG) was observed significantly more in fetuses included in the control group (Table 2).

### 3.3. Overall Fetal Outcomes

The overall neonatal mortality rate was 2.0%, and there was no significant statistical difference between the two study groups (*p* = 0.19). In addition, no significant difference was observed between the rates of grades I–II intraventricular hemorrhages (IVH) in the two groups (*p* = 0.41). However, bronchopulmonary dysplasia and grades III–IV intraventricular hemorrhages were more frequently found in participants included in group two (control) (Table 3).

### 3.4. Fetal Outcomes According to the Time of Delivery, in Infants Born at Less than 30 Weeks of Gestation

Gestational age was a major factor in the occurrence of neonatal complications in infants with FGR, with the majority of complications being reported in those born before 30 WG.

Note that the difference between the groups was the time of delivery (according to Doppler indexes) in infants with a gestational age less than 30 WG. 

In infants delivered up to 30 WG, grades III/IV intraventricular hemorrhages and bronchopulmonary dysplasia were statistically significantly more frequent in the control group. On the contrary, grades I/II intraventricular hemorrhages were more frequent in fetuses whose delivery time was the onset of UA AEDF/REDF (Table 4).

Univariate binomial logistic regression analysis on fetuses born under 30 WG shows that those included in the control group were 30 times more likely to develop bronchopulmonary dysplasia (*p* = 0.002) and 14 times more likely to develop IVH grades III/IV (*p* < 0.001) (Table 5).

Additionally, variables including age, abnormal CTG, and abnormal BPP may be correlated with poor neonatal outcomes. Gestational age and frequency of abnormal BPP were similar between the two groups, whereas a pattern of abnormal CTG was observed more frequently in subjects included in the control group (3.2% vs. 27.8%; *p* = 0.02) (Table 6).

Therefore, the differences, in terms of the likelihood of developing grades III/IV IVH and bronchopulmonary dysplasia, remain significantly higher in the control group and when adjusting for gestational age at birth (Table 7). 

Furthermore, multivariate logistic regression, including gestational age, CTG, and abnormal BPP, shows that bronchopulmonary dysplasia and grades III/IV intraventricular hemorrhage are 23 and 70 times more likely in control newborns, respectively (Table 8).

### 3.5. Fetal Outcomes According to the Time of Delivery, in Infants Born between 30 and 34 Weeks of Gestation

A total of 120 newborns were delivered between 30 and 34 WG. Among them, there were no neonatal deaths or infants who developed bronchopulmonary dysplasia and necrotizing enterocolitis. Instead, 31 newborns developed intraventricular hemorrhage, 29 of them grades I/II and 2 infants grades III/IV. Grades I/IV intraventricular hemorrhage was statistically significantly more common in newborns delivered on the occurrence of absent end-diastolic flow on UA (*p* < 0.0001). Furthermore, grades III/IV intraventricular hemorrhage occurred only among newborns included in the control group (delivered after the occurrence of reversed end-diastolic flow on the UA) (Table 9).

Univariate logistic regression showed that the likelihood of developing grades I/II intraventricular hemorrhage was 7.76 times higher among newborns whose delivery was delayed until the REDF on the UA compared with those born at the time of the onset of UA-AEDF (Table 10).

Furthermore, multivariate logistic regression showed a 14-fold higher odds of developing IVH I/II in infants included in the control group, adjusted for gestational age (Table 11).

## 4. Discussion

### 4.1. Key Results

In a cohort of pregnancies complicated by early onset FGR, we observed that early neonatal complications were associated with the timing of delivery in relation to Doppler parameters. 

Infants under 30 weeks were examined separately due to the fact that among them, in the control group, birth was decided according to Doppler parameters of the ductus venosus. Typically, umbilical artery Doppler is not included in management protocols for early onset FGR, up to 30–32 WG. Instead, the clinician relies on other indicators of fetal health, such as fetal heart rate (FHR) tracing, biophysical profile, and Doppler velocimetry of the ductus venosus [10,11]. In contrast, PI ductus venosus and short-term variation in fetal heart rate are known to be important predictors of optimal birth timing before 32 weeks gestation and correlate with fetal outcome at birth in certain studies [12].

Firstly, among infants born under 30 weeks of gestation in whom delivery was delayed until the appearance of the absent/reversed DV wave, the likelihood of BPD was 30-fold higher compared with infants born at the time of UA-AEDF/REDF. Furthermore, grade IVH grades III/IV was significantly more common among infants whose delivery was delayed to the occurrence of reversed/absent DV a-wave compared with those born immediately after the occurrence of an umbilical artery flow anomaly. In addition, the likelihood of an increased degree of intraventricular hemorrhage was 14 times higher among infants in whom birth was delayed until the time of absent/reversed DV a-wave.

Secondly, the study also assessed the likelihood of neonatal complications in fetuses over 30 WG delivered at the onset of UA-AEDF (case group) compared with those whose birth was delayed until the onset of UA-REDF (control group). Infants whose delivery was delayed until the onset of UA-REDF were 7.76 times more likely to develop grades I/II intraventricular hemorrhage compared with those born at the time of the onset of UA-AEDF.

Overall, in our study, neonatal mortality was 4%, while 4.9% of newborns developed bronchopulmonary dysplasia and 39% intraventricular hemorrhage of varying degrees.

### 4.2. Interpretation 

In terms of neonatal complications of FGR, a systematic review showed an overall prenatal death rate of 12.3% and a neonatal mortality rate among FGR fetuses of 6.6% [1]. The randomized umbilical and fetal flow study in Europe (the TRUFFLE study) also reported a perinatal mortality rate of 8%, significantly higher than that reported in our study [13]. 

Regarding neonatal complications, one study found that at 27 weeks GA, 25% of non-FGR infants developed BPD, whereas 60% of infants with moderate FGR and 90% of infants with severe FGR developed BPD [14]. In our cohort, the overall rate of BPD was significantly lower compared with these results, being found only in infants born under 30 WG. However, in neonates in whom delivery was delayed until the onset of absent/reversed DV a-wave, the incidence of BPD was comparable to that reported in previous studies. Despite the fact that birth weight and gestational age are inversely correlated with the incidence of BPD, the results of our research indicate an increased risk of BPD in children with FGR whose birth was delayed after the development of umbilical artery Doppler anomalies [15]. These results suggest that FGR severity is a greater determinant of BPD than gestational age itself. According to studies, the biological mechanisms that cause FGR also lead to vulnerability in developing lungs [15]. 

Further, a major complication of prematurity is IVH, with our results showing that all FGR infants delivered under 30 WG developed it to varying degrees. Additionally, infants whose birth was delayed until the onset of absent/inverted DV a-wave were significantly more likely to develop BPD and severe degrees of IVH. However, grades I/II intraventricular hemorrhage was more common among infants born earlier at the time of UA-AEDF/REDF onset. We interpret these results to mean that the most significant contributor to IVH is extreme prematurity. This explains why all infants in our study born at GA < 30 GW developed this complication. However, the increased risk of high-grade IVH in infants whose birth was delayed may indicate that the severity of FGR influences the degree of IVH. Other studies have also shown that the loss of diastolic velocity of the umbilical artery significantly increases the risk of neonatal intraventricular hemorrhage. However, prematurity remains the most important factor of intraventricular hemorrhage [16]. Moreover, previous studies have shown that in preterm infants with FGR, IVH was more common compared with preterm infants with appropriate growth [17]. 

In addition, A. Valcamonico et al. showed that FGR infants with absent or reversed end-diastolic flow in the umbilical arteries have, in addition to increased fetal and neonatal mortality, a higher incidence of long-term permanent neurological damage compared with fetuses with growth delays with the diastolic flow in the umbilical circulation [18]. 

### 4.3. Limitations

This study has several limitations. First, the study data were obtained from a single clinic. Second, the sample may not have been large enough to assess the likelihood of neonatal death, as only four deaths were included in the analysis in this cohort. Furthermore, the study only evaluates early neonatal complications, without assessing long-term consequences.

### 4.4. Clinical Implications

There is no effective antenatal treatment for placental dysfunction, and therefore once FGR has been identified, the principal management steps are the institution of fetal surveillance and determination of appropriate thresholds for delivery. The antenatal detection rates of FGR are estimated to be between 25% and 36% [19]. There is a consensus regarding the diagnosis of early onset FGR. Early FGR is defined as the presence at GA < 32 weeks, in the absence of congenital anomalies, of the following: AC/EFW < 3rd centile or UA-AEDF; or AC/EFW < 10th centile combined with UtA-PI > 95th centile and/or UA-PI > 95th centile [20]. 

Even if there is a consensus on the diagnosis, there are different opinions on the management. Therefore, more studies on neonatal complications by time of birth are needed to determine a current internationally accepted consensus. The risk of fetal degeneration and stillbirth in pregnancies under surveillance versus neonatal morbidity and mortality related to preterm delivery is the main management challenge. 

Delaying the delivery of a growth-restricted preterm fetus with the aim of extending gestation should be the main management objective, according to some observational and randomized studies [21,22,23]. From 24 to 28 weeks of gestation, each day of pregnancy prolongation results in an estimated 2% decrease in neonatal death, as well as major neonatal complications including bronchopulmonary dysplasia, high-grade intraventricular hemorrhage, and surgical necrotizing enterocolitis. The impact of prematurity, low and extremely low neonatal weights, difficult resuscitation and low Apgar scores, and absent/reversed DV a-wave have been suggested as risk factors for mortality in preterm infants [24,25,26]. Additionally, there is a strong correlation between newborn growth restriction and the risk of neonatal death [27]. Therefore, when early onset FGR is diagnosed, the timing of the delivery decision becomes critical.

Delivery standards for early onset FGR that manifests between 26 and 32 weeks of gestation have been established by the TRUFFLE RCT. An absent or reversed DV a-wave, the cCTG safety criteria, or an abnormal biophysical profile score of four or less are indications for delivery during this gestational age window [21]. The TRUFLE trial evaluated two monitoring strategies and specific delivery criteria in infants with early onset FGR, with survival without neurodevelopmental impairment at 2 years of age as the primary outcome. The strategy of waiting for the absence or reversal of the DV a-wave to determine delivery produced the best results in the study. The stillbirth rate did, however, quadruple when compared with patients who underwent cCTG and umbilical artery Doppler monitoring [28,29]. 

However, according to the RCOG, the use of umbilical artery Doppler in a high-risk population has been shown to reduce perinatal morbidity and mortality and should be the primary surveillance tool in the SGA fetus [30]. Umbilical artery Doppler monitoring should be initiated when the fetus is considered viable and FGR is suspected. Some recommendations indicate that although Doppler studies of the ductus venosus, middle cerebral artery, and other vessels have some prognostic value as part of the assessment of fetal well-being in pregnancy, these should be reserved for research protocols [31]. 

Our study shows the benefit of delivering fetuses before the onset of absent/reversed DV-a wave, the risk of early neonatal complications being higher among infants with GA < 30 WG whose delivery was delayed after the onset of UA-AEDF/REDF. Furthermore, in recent years, the capacity for the neonatal care of premature infants has improved. In particular, over the past 50 years, the outlook for infants with a birth weight of approximately 1 kg has changed from 95% mortality in 1960 to 95% survival in 2000 [32]. 

### 4.5. Research Implications

These results can be used to develop a new strategy for the management of early onset FGR. Contrary to the TRUFFLE study indications, waiting until the occurrence of absent/reversed DV a-wave leads to a higher probability of neonatal complications. Future studies should confirm these results, as well as determine long-term outcomes.

## 5. Conclusions

Newborns with GA < 30 WG delivered when UA-AEDF/REDF is present, before the onset of absent/reversed DV a-wave, are less likely to develop grades III/IV intraventricular hemorrhage and bronchopulmonary dysplasia. Given the remarkable improvement in neonatal care capacity of preterm infants, we conclude that the appropriate time for delivery of GA < 30 GW infants with early onset FGR is at the occurrence of UA-AEDF/REDF. In addition, in FGR infants with GA > 30 WG, delivery at the occurrence of UA-AEDF is more beneficial in terms of decreased risk of early neonatal complications compared with infants whose birth was delayed until the occurrence of UA-REDF.

## Figures and Tables

**Table 1 jpm-13-00645-t001:** Characteristics of 205 participants with early onset FGR.

Parameters	Overall	Group 1 (Cases) n = 120	Group 2 (Control) n = 85	*p* Value
Age [mean ± SD]	29.48 ± 5.44	29.31 ± 5.34	29.72 ± 5.61	0.59
Height [median (IQR)]	162 (8)	162 (9)	162 (8)	1.00
Weight [median (IQR)]	74 (14)	73 (14)	74 (14)	0.58
Gravidity [median (IQR)]	1 (1)	1 (1)	1 (1)	0.83
Parity [median (IQR)]	0 (1)	0 (1)	0 (1)	0.72
WG on diagnosis [median (IQR)]	31 (2)	31 (2)	31 (2)	0.75
WG on delivery [median (IQR)]	32 (2)	32 (3)	33 (3)	0.12

Note: WG = weeks of gestation.

**Table 2 jpm-13-00645-t002:** Doppler findings in 205 early onset FGR fetuses.

Parameters	Overall	Group 1 (Case)	Group 2 (Control)	*p* Value
UA AEDF	115/56.1%	90/75.0%	25/29.4%	<0.001
UA REDF	69/33.7%	30/25.0%	39/45.9%	0.002
DVA absent	14/6.8%	-	14/16.5%	NA
Reversed DVa	7/3.4%	-	7/8.2%	NA
Abnormal CTG	9/4.4%	1/0.8%	8/9.4%	0.004
Abnormal BPP	5/2.4%	1/0.8%	4/4.7%	0.09

Note: AEDF = absent end-diastolic flow; BPP = biophysical profile; CTG = cardiotocography; REDF = reversed end-diastolic flow; UA = umbilical artery.

**Table 3 jpm-13-00645-t003:** Fetal outcomes in 205 early onset FGR fetuses.

Parameters	Overall	Case Group	Control Group	*p* Value
Neonatal death	4/2.0%	1/0.8%	3/3.5%	0.19
Bronchopulmonary dysplasia	10/4.9%	1/0.8%	9/10.6%	0.002
IVH grades I/II	55/26.8%	29/24.2%	24/28.2%	0.52
IVH grades III/IV	25/12.2%	8/6.7%	17/20.0%	0.004
NEC	2/1.0%	-	2/2.4%	NA

Note: IVH = intraventricular hemorrhage. NEC = Necrotizing enterocolitis.

**Table 4 jpm-13-00645-t004:** Adverse fetal outcomes in infants born at less than 30 weeks.

Parameters	Total (n = 49)	Case Group (Delivery According to UA) (n = 31)	Control Group (Delivery According to DV) (n = 18)	*p* Value
Neonatal death	4/8.2%	1/3.2%	3/16.7%%	0.13
Bronchopulmonary dysplasia	10/4.9%	1/3.2%	9/50.0%	<0.001
IVH I/II	25/51.0%	23/74.2%	2/11.1%	<0.001
IVH III/IV	23 12.2%	8/25.8%	15/83.3%	<0.001
NEC	2/4.1%	-	2/11.1%	NA

IVH = intraventricular hemorrhage. NEC = Necrotizing enterocolitis.

**Table 5 jpm-13-00645-t005:** The likelihood of early onset FGR infants, under 30 WG, included in the control group to develop adverse outcomes.

Outcomes	B	S.E	*p* Value	Odds Ratio	95% CI	
					Lower	Upper
Neonatal death	1.79	1.19	0.13	6.00	0.574	62.69
Bronchopulmonary dysplasia	3.40	1.12	0.002	30.00	3.33	269.7
IVH grades I/II	−3.135	0.85	<0.001	0.043	0.008	0.23
IVH grades III/IV	2.665	0.754	<0.001	14.37	3.280	63.0

Note: The reported data represent the likelihood of complications in the control group compared with the case group; IVH = intraventricular hemorrhage.

**Table 6 jpm-13-00645-t006:** Descriptive statistics of confounding variables among 45 infants born under 30 weeks of gestation.

Variable	Total (n = 49)	Case Group (Delivery According to UA) (n = 31)	Control Group (Delivery According to DV) (n = 18)	*p* Value
GA on delivery	29 (2)	29 (2)	28 (1)	0.23
Abnormal CTG	6/12.2%	1/3.2%	5/27.8%	0.02
Abnormal BPP	4/8.2%	1/3.2%	3/16.7%	0.13

BPP = biophysical profile; CTG = cardiotocography; GA = gestational age.

**Table 7 jpm-13-00645-t007:** Probability of fetal outcomes among infants delivered under 30 weeks in the control group, after adjustment for gestational age at birth.

Outcomes	B	S.E	*p* Value	Adjusted Odds Ratio	95% CI	
					Lower	Upper
Bronchopulmonary dysplasia	3.40	1.15	0.003	30.24	3.126	292.6
IVH I/II	−3.836	1.113	0.001	0.02	0.002	0.191
IVH III/IV	3.186	0.968	0.001	24.195	3.628	161.350

Note: The reported data represent the likelihood of complications in the control group, adjusted for GA, compared with the case group.

**Table 8 jpm-13-00645-t008:** Likelihood of fetal outcomes among infants delivered under 30 WG in the control group, on multivariate analysis.

Outcomes	B	S.E	*p* Value	Adjusted Odds Ratio	95% CI	
					Lower	Upper
Bronchopulmonary dysplasia	3.14	1.19	0.008	23.16	2.25	238.4
IVH I/II	−4.820	1.49	0.001	0.08	0.00	0.150
IVH III/IV	4.253	1.36	0.002	70.29	4.86	1016.64

Note: The reported data represent the likelihood of complications in the control group compared with the case group. Confounding variables = “gestational weeks”, “abnormal CTG, abnormal BPP”.

**Table 9 jpm-13-00645-t009:** Adverse fetal outcomes in infants born between 30 and 34 WG.

Parameters	Total (n = 120)	Case Group [Delivery at the Advent of the UA-AEDF] (n = 76)	Control Group- [Delivery Delayed Until UA-REDF] (n = 44)	*p* Value
IVH I/II	29/24.2%	8/10.5%	21/47.7%	<0.0001
IVH III/IV	2/1.7%	-	2/4.5%	NA

IVH = intraventricular hemorrhage, NA = not applicable.

**Table 10 jpm-13-00645-t010:** The likelihood of infants born between 30 and 34 WG included in the control group to develop intraventricular hemorrhage grades I–II.

Outcomes	*p* Value	Odds Ratio	95% CI	
			Lower	Upper
IVH grades I/II	<0.001	7.76	3.02	19.9

Note: The reported data represent the likelihood of IVH grades I/II in the control group compared with the case group; IVH = intraventricular hemorrhage.

**Table 11 jpm-13-00645-t011:** Likelihood of intraventricular hemorrhage grades I–II among infants delivered between 30 and 34 weeks in the control group on multivariate analysis.

Models	*p* Value	Adjusted Odds Ratio	95% CI	
			Lower	Upper
Model 1	<0.001	14.37	4.32	47.76
Model 2	<0.001	14.49	4.36	48.14

Model 1: The reported data represent the likelihood of intraventricular hemorrhage grades I–II in the control group, adjusted for GA, compared with the case group. Model 2: The reported data represent the likelihood of intraventricular hemorrhage grades I–II in the control group, adjusted for GA, presence of abnormal CTG, and presence of abnormal BPP.

## Data Availability

The data sets used and/or analyzed during the present study are available from the correspondence author on reasonable request.

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
