# Peer review of "Appropriate Delivery Timing in Fetuses with Fetal Growth Restriction to Reduce Neonatal Complications: A Case—Control Study in Romania"

_jpm, 2023, doi:10.3390/jpm13040645_

Round 1

Reviewer 1 Report

Thank you for giving this opportunity to me.

The authors conducted a well-known challenge in the field of perinatal medicine regarding the timing of delivery in intrauterine growth restriction fetuses. The groups compared are well designed and the appropriate statistical analysis applied. It has originality from this point of view. Before publications, i would suggest some minor corrections as follows:

1) some spelling errors should be corrected such as line 270 '' iIn our study...''

2) There are some studies that evaluated abnormal doppler parameter in IUGR fetuses with some ischemic markers. one-two sentences shall be added to the discussion section regarding this issue. Please refer this study (doi: 10.1080/14767058.2019.1569623.)

Author Response

RESPONSE TO REVIEWER 1

1) some spelling errors should be corrected such as line 270 '' iIn our study...''

Response: Thank you for your comment. We have corrected it.

2) There are some studies that evaluated abnormal doppler parameter in IUGR fetuses with some ischemic markers. one-two sentences shall be added to the discussion section regarding this issue. Please refer this study (doi: 10.1080/14767058.2019.1569623.)

Response: Thank you for the suggestion. We have added references from your proposed manuscript to the discussion.

Reviewer 2 Report

I am pleased to have the possibility to review the study “Appropriate Delivery Timing in Fetuses with Fetal Growth Restriction to Reduce Neonatal Complications: A Case-Control Study in Romania”. The discussed problem of fetal growth restriction has a meaningful impact on maternal and children’s health. The size of the study population and the statistical methods used are undoubtful study strengths. Nevertheless, several methodological issues require clarification before publication. 

Major points:

1.     The aim of the study is not reached. The authors attempt to estimate the pregnancy timing. Nor calculations were made in this direction, nor was the problem discussed. In conclusion, another aim of the study was presented. I strongly suggest reviewing the survey after estimating and examining the study's objective.

2.     The methodology is well conducted. Strong analytic methods were used since the multivariate regression model was performed to estimate the risk factors of intraventricular haemorrhage. Nevertheless, strong analytical methods not appropriately applied to the study objective provide the readers with any conclusion.

3.     It is not presented what inclusion criteria were used to allocate patients into the case and control groups. Please make it more precise and more appropriate to the aim of the study.

4.     I strongly suggest using the STROBE guidelines for cohort studies from the EQUATOR website to improve the quality of the research and, first of all, in the discussion section. There are no limitations of the study presented, and the first paragraph of the discussion is no summary of results but the introduction section. Moreover, the clinical application of the study must be more strongly highlighted in the study. 

5.     I am also not fully agreeing with the authors about lacking consensus on the timing of the delivery of FGR fetuses. I strongly recommend and probably compare your study to the following papers with doi: 

10.1002/uog.15884

10.1080/14767058.2019.1676412

10.1016/j.bpobgyn.2018.02.009

10.3390/children8060522

10.1002/ijgo.13522

6.     It is also not presented what was inclusion criteria according to the time of diagnosis and time of delivery. Was it the inclusion/exclusion criterium at all? You write in line 99 that it was 32 weeks of gestation, and in lines 108 and 112, you write that 34 weeks of gestation were also included, and the median gestational age written in line 145 was 32. It could not be possible by the inclusion of fewer than 32 weeks of gestation pregnancies. This part of the study is unclear (lines 98-118).

7.     Why you excluded preeclampsia, HELLP and pathological CTG? Please explain it in the discussion section or direct in the methodology.

8.     Why were fetuses under 30 weeks of gestation separately examined? Please make appropriate clarifications in methodology or discussion. 

Minor points:

1.     Language (grammar and style) revision of the manuscript of the native speaker is required.

2.     Definition, including the timing of diagnosis, is required in the introduction.

3.     FGR and IUGR (line 154) were used as synonyms in this study. This is wrong. Only the FGR definition should be considered, according to novel guidelines.

4.     It should be “weeks of gestation” instead of “GW” or “gestational week”.

5.     What means “common disorder” (line 48)? 2% of pregnancies are not so common. Please be correct.

In my opinion, the discussed problem of early onset FGR is very important, underestimated and has a meaningful impact on maternal and children’s health, as the authors strongly suggest. The quality of the study is high, but several methodological issues should be improved. The study must be carefully revised, and the authors should reconsider the aim of the study before resubmission. 

Author Response

RESPONSE TO REVIEWER 2

Major points:

  1. The aim of the study is not reached. The authors attempt to estimate the pregnancy timing. Nor calculations were made in this direction, nor was the problem discussed. In conclusion, another aim of the study was presented. I strongly suggest reviewing the survey after estimating and examining the study's objective.

Response: Thank you for your comment. Indeed, we did not conclude what is the right time to give birth to fetuses with FGR. The wording in the aim statement was wrong. This study was conducted to determine the risk of neonatal complications in fetuses with delayed delivery. Therefore, this study achieved its aim and demonstrated that the likelihood of developing a neonatal complication is lower in fetuses with FGR born earlier (up to ductus venosus anomaly/end-diastolic flow reversal on UA) despite a lower gestational age.

  1. The methodology is well conducted. Strong analytic methods were used since the multivariate regression model was performed to estimate the risk factors of intraventricular haemorrhage. Nevertheless, strong analytical methods not appropriately applied to the study objective provide the readers with any conclusion.

Thank you for your comment. As stated in the above paragraph, this study was designed to determine probabilities. The construction of the database and the statistical analysis was done from the beginning for this purpose. As stated above, the statement of purpose was an error of expression which we have corrected.

  1. It is not presented what inclusion criteria were used to allocate patients into the case and control groups. Please make it more precise and more appropriate to the aim of the study.

Thank you. We have modified it to be clearer.

  1. I strongly suggest using the STROBE guidelines for cohort studies from the EQUATOR website to improve the quality of the research and, first of all, in the discussion section. There are no limitations of the study presented, and the first paragraph of the discussion is no summary of results but the introduction section. Moreover, the clinical application of the study must be more strongly highlighted in the study. 

Thank you. We have adjusted the discussions according to the STROBE guidelines.

  1. I am also not fully agreeing with the authors about lacking consensus on the timing of the delivery of FGR fetuses. I strongly recommend and probably compare your study to the following papers with doi: 

10.1002/uog.15884

10.1080/14767058.2019.1676412

10.1016/j.bpobgyn.2018.02.009

10.3390/children8060522

10.1002/ijgo.13522

Response: Thank you for your comment. We have included your proposed manuscripts in the discussion.

  1. It is also not presented what was inclusion criteria according to the time of diagnosis and time of delivery. Was it the inclusion/exclusion criterium at all? You write in line 99 that it was 32 weeks of gestation, and in lines 108 and 112, you write that 34 weeks of gestation were also included, and the median gestational age written in line 145 was 32. It could not be possible by the inclusion of fewer than 32 weeks of gestation pregnancies. This part of the study is unclear (lines 98-118).

Response: Thank you for your comment.  In our methods we did not write anywhere that only fetuses under 32 weeks were included. The main criterion for inclusion was that the fetus was diagnosed with early onset FGR. This diagnosis was established as explained in line 99. One point of the definition of early-onset FGR is that the diagnosis (not the birth) is established at gestational age below 32 weeks. In summary, we have shown that fetuses were included that were diagnosed with FGR at GA<32 weeks (at diagnosis). They were followed up and delivered according to the group to which they belonged.

  1. Why you excluded preeclampsia, HELLP and pathological CTG? Please explain it in the discussion section or direct in the methodology.

Response: These cases required emergency delivery independent of Doppler parameters. We have included this in the methodology.

  1. Why were fetuses under 30 weeks of gestation separately examined? Please make appropriate clarifications in methodology or discussion. 

Infants under 30 weeks were examined separately due to the fact that among them, in the control group, birth was decided according to Doppler parameters of the ductus venosus.

Minor points:

  1. Language (grammar and style) revision of the manuscript of the native speaker is required.

Response: Thank you. The manuscript has been checked by a native English speaker

  1. Definition, including the timing of diagnosis, is required in the introduction.

Response: Thank you. However, the definition and timing of the diagnosis is set out on lines 50-53.

  1. FGR and IUGR (line 154) were used as synonyms in this study. This is wrong. Only the FGR definition should be considered, according to novel guidelines.

Response: Thank you. We have corrected it.

  1. It should be “weeks of gestation” instead of “GW” or “gestational week”.

Response: hank You. We have revised

  1. What means “common disorder” (line 48)? 2% of pregnancies are not so common. Please be correct.

Response: Thank You. We have revised

Round 2

Reviewer 2 Report

Thank you very much for included changes. They are improving your work a lot. Nevertheless, several minor changes should be clarified before publication. 

1.     I will recommend including in the methodology section a sentence about STROBE guidelines. For example, “This study was performed according to the STROBE guidelines” + citation.

2.     “Infants under 30 weeks were examined separately due to the fact that among them, in the control group, birth was decided according to Doppler parameters of the ductus venosus.” – do you include this information in the discussion? It is important to provide the readers with such information. 

I mean the explanation of the fact why in fetuses under 30 weeks of gestation, the ductus v. doppler is so important. The comparison with other studies will also increase the value of your statement.

Discussion

3.     I would recommend avoiding the “numbers” in this section, like “conducted on 205 infants diagnosed with early-onset FGR” – you have shown this in your results. Try to interpret this information. 

4.     As well the aim of the study was given in the introduction. Statements like “the study evaluated the likelihood of” or “the study aimed” are redundant, and you don’t have to repeat them. It is not false, but the answer to your research question is more important.

5.     I recommend beginning this section with the most relevant message you want to send to the readers, like “…. infants whose birth was delayed until the onset of absent/reversed DV-a wave were significantly more likely to develop BPD and severe grades of IVH“.

6.     The sentence in line 284 is not finished “However, …”

7.     I will also no recommend starting the discussion with limitations of the study. Include them close to the end of the discussion before describing the clinical implementations of your results.

8.     Is the sentence “Therefore, participants of this study born under 30 WG were examined separately from those born after 30 …” the most important result of your study?

9.     Information in paragraph 5 in lines 293-300 includes information usually showed in the introduction. As well as lines 276-278, ……

10.  Line 305: “In contrast, in our study, the rate of BPD was 4.9%.” – please discuss why it is so.

11.  “For instance, in a study of the Polish population, the Polish growth chart was used to define a total of 9.8% SGA, which was higher than the percentages found using the Fenton and Intergrowth Project charts[17].”  - the most important message from this study is necessary to develop the population-based growth charts to improve the neonatal outcomes.

12.  “Even if there is a consensus on the diagnosis, there are different opinions on the management” – what do you propose? 

13.  “Delaying the delivery of the growth-restricted preterm fetus with the aim of extending gestation should be the main management objective, according to some observational and randomised studies” – it is a very strong suggestion. Off course delaying the delivery would be good, but we have to think about the consequences of placenta issuing in the FGR fetuses and, according to the best knowledge and individual assessment of every case, to decide when the timing for delivery is, what is proposed in citated consensus. 

14.  “Compared to an SGA-only relationship, this one is much more substantial” – I don’t understand what the authors mean.

15.  “to 95% survival in 2000” – not only survival matters. It is important to pay attention to the further neurodevelopment of neonates born with a weight of around 1 000 g. 

16.  “Furthermore, in our study, the incidence of neonatal complications was higher among infants whose delivery was delayed after the occurrence of UA-AEDF.” – how long have you been waiting and what indicators were in these fetuses for ending the pregnancy? 

17.  “As there is not yet a consensus on the management of early-onset FGR, our aim was to determine the risk of short-term neonatal complications based on the timing of delivery and Doppler indices, respectively.” – it is not a conclusion. 

Author Response

Thank you for your time in conducting this review. Your comments have helped us to make major improvements to the manuscript.

Below are our responses:

  1. I will recommend including in the methodology section a sentence about STROBE guidelines. For example, “This study was performed according to the STROBE guidelines” + citation.

Response: Thank you for the suggestion. We have added this.

  1. “Infants under 30 weeks were examined separately due to the fact that among them, in the control group, birth was decided according to Doppler parameters of the ductus venosus.” – do you include this information in the discussion? It is important to provide the readers with such information.I mean the explanation of the fact why in fetuses under 30 weeks of gestation, the ductus v. doppler is so important. The comparison with other studies will also increase the value of your statement.

Response: Thank you for the suggestion. We have included this statement in discussion-paragraph 2.

  1. I would recommend avoiding the “numbers” in this section, like “conducted on 205 infants diagnosed with early-onset FGR” – you have shown this in your results. Try to interpret this information.

Response: Thank you for the suggestion. At the request of the academic editor we have restructured the discussions according to the STROBE guidelines. We have tried to eliminate as much as possible the repetition of results in the discussion section. However, according to the STROBE guidelines we had to present the key results in the discussion.

  1. As well the aim of the study was given in the introduction. Statements like “the study evaluated the likelihood of” or “the study aimed” are redundant, and you don’t have to repeat them. It is not false, but the answer to your research question is more important.

Thank you for the suggestion. We have removed these phrases from the discussion section

  1. I recommend beginning this section with the most relevant message you want to send to the readers, like “…. infants whose birth was delayed until the onset of absent/reversed DV-a wave were significantly more likely to develop BPD and severe grades of IVH“.

Response: Thank you for the suggestion. This message is being provided in the key results subsection at the beginning of the discussion section.

  1. The sentence in line 284 is not finished “However, …”

Response:  Thank you. It was a typo. We have removed

  1. I will also no recommend starting the discussion with limitations of the study. Include them close to the end of the discussion before describing the clinical implementations of your results.

Thank you for the suggestion. We have moved the limitations of the study forward to clinical implications.

  1. Is the sentence “Therefore, participants of this study born under 30 WG were examined separately from those born after 30 …” the most important result of your study?

Response: We have removed this line

  1. Information in paragraph 5 in lines 293-300 includes information usually showed in the introduction. As well as lines 276-278, ……

Response: Thank you for the suggestion, we have deleted the phrase from lines 293-300. However, the statement in line 278 we consider necessary to justify the splitting of cases, as you requested in point 2.

  1. Line 305: “In contrast, in our study, the rate of BPD was 4.9%.” – please discuss why it is so.

Response: Thank you for the suggestion. Since at the request of the academic editor we have restructured the discussion this statement is no longer in line 305, but in the subsection "key results". We have also added discussion on this issue in the subsection "Interpretation".

  1. “For instance, in a study of the Polish population, the Polish growth chart was used to define a total of 9.8% SGA, which was higher than the percentages found using the Fenton and Intergrowth Project charts[17].” - the most important message from this study is necessary to develop the population-based growth charts to improve the neonatal outcomes.

Response: Thank you for the comment. We have added this.

  1. “Even if there is a consensus on the diagnosis, there are different opinions on the management” – what do you propose?

Response: We believe that more studies on neonatal complications by time of birth are needed to determine a suitable current internationally agreed consensus. We have added this to the discussions.

  1. “Delaying the delivery of the growth-restricted preterm fetus with the aim of extending gestation should be the main management objective, according to some observational and randomised studies” – it is a very strong suggestion. Off course delaying the delivery would be good, but we have to think about the consequences of placenta issuing in the FGR fetuses and, according to the best knowledge and individual assessment of every case, to decide when the timing for delivery is, what is proposed in citated consensus.

Response: We are in complete agreement with you.  However as we have pointed out in discussion some recommendations are to wait for late Doppler DV changes to determine delivery. These practices seem to have poor neonatal outcomes both in our study and as we have observed in clinical practice.

  1. “Compared to an SGA-only relationship, this one is much more substantial” – I don’t understand what the authors mean.

Response: We deleted that statement

  1. “to 95% survival in 2000” – not only survival matters. It is important to pay attention to the further neurodevelopment of neonates born with a weight of around 1 000 g.

Response: We are aware of this. We also included in the limitations of the study the fact that newborns were not followed up for long-term sequelae. We also stated in the discussion that future studies should also determine long-term outcomes according to the time of birth. Concerning the neonatal care evolution we did not find in the literature data on the long-term complication rate trends.

  1. “Furthermore, in our study, the incidence of neonatal complications was higher among infants whose delivery was delayed after the occurrence of UA-AEDF.” – how long have you been waiting and what indicators were in these fetuses for ending the pregnancy?

Response: We have restructured the discussions this phrase being clearer. In fetuses with GA>30WG we indicated birth at the appearance of UA-AEDF (group of cases) or waited until the appearance of UA-REDF (as explained in the methods).

  1. “As there is not yet a consensus on the management of early-onset FGR, our aim was to determine the risk of short-term neonatal complications based on the timing of delivery and Doppler indices, respectively.” – it is not a conclusion.

Response: Thank you for your comment. We have deleted this sentence.